# Accelerating Linear Recurrent Neural Networks for the Edge with Unstructured Sparsity

Alessandro Pierro [* 1 2]   Steven Abreu [* 1 3]   Jonathan Timcheck [1]   Philipp Stratmann [1]   Andreas Wild [1]
Sumit Bam Shrestha [1]

## Abstract

Linear recurrent neural networks enable powerful long-range sequence modeling with constant memory usage and time-per-token during inference. These architectures hold promise for streaming applications at the edge, but deployment in resource-constrained environments requires hardware-aware optimizations to minimize latency and energy consumption. Unstructured sparsity offers a compelling solution–when accelerated by compatible hardware platforms. In this paper, we investigate the Pareto front of performance and efficiency across inference compute budgets. We find that highly sparse linear RNNs consistently achieve better efficiency-performance trade-offs than dense baselines, with $2\times$ less compute and $36\%$ less memory iso-accuracy. Our models achieve state-of-the-art results on a streaming audio denoising task. By quantizing our sparse models to fixed-point arithmetic and deploying them on the Intel Loihi 2 neuromorphic chip, we translate model compression into tangible gains of $42\times$ lower latency and $149\times$ lower energy consumption compared to a dense model on an edge GPU. Our findings showcase the transformative potential of unstructured sparsity, paving the way for highly efficient recurrent neural networks in real-world, resource-constrained environments.

 https://github.com/IntelLabs/SparseRNNs

## 1. Introduction

Linear Recurrent Neural Networks (RNNs) have recently emerged as powerful primitives for sequence modeling, both in isolation or hybridized with self-attention, achieving impressive results in language modeling (Poli et al., 2024), audio generation (Goel et al., 2022), and genomics (Nguyen et al., 2023), and many other areas. This success has been ignited by advances in initialization, parametrization, and parallelization of these models, which, combined, enabled large-scale training on GPUs (Voelker et al., 2019; Chilkuri et al., 2021; Gu et al., 2020; 2022b; Smith et al., 2023).

At inference time, linear RNNs iteratively compress the input sequence into a finite-dimensional representation whose dimensionality does not depend on the sequence length. Their memory requirements remain constant regardless of sequence length, and runtime scales linearly with sequence length. In contrast, transformer architectures (Vaswani et al.,

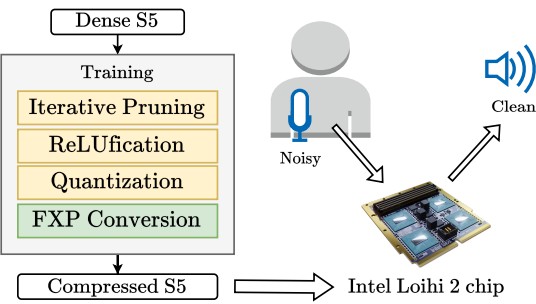

*Figure 1.* Model compression and acceleration pipeline for linear RNNs, tailored to the Intel Loihi 2 chip.

2017) exhibit linear memory growth and quadratic runtime scaling as sequence length increases. This advantageous scaling makes linear RNNs especially well-suited for real-time long-range sequence modeling on edge devices that require low latency, a small form factor, and are subject to weight and power constraints, as common for applications like audio denoising (Timcheck et al., 2023), keyword spotting (Warden, 2018), or perception-and-control (Lu et al., 2023). Although model optimization and compression are essential for enabling efficient edge machine learning by reducing resource demands, their application to accelerate the inference of linear RNNs remains under-explored.

---

[*]Equal contribution  [1]Neuromorphic Computing Lab, Intel Corporation, USA [2]Institute of Informatics, LMU Munich, Germany [3]Bernoulli Institute & CogniGron, University of Groningen, Netherlands. Correspondence to: Alessandro Pierro <alessandro.pierro@intel.com>.

*Proceedings of the $42^{nd}$ International Conference on Machine Learning*, Vancouver, Canada. PMLR 267, 2025. Copyright 2025 by the author(s).

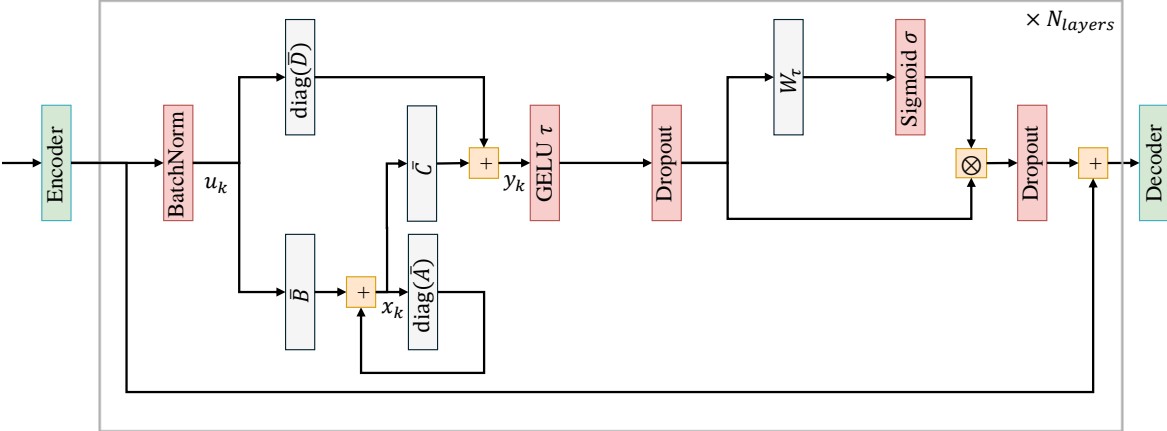

*Figure 2.* Overview of the S5 architecture. Symbols are shown as defined by equations in Section 2.1.

Linear RNNs are a promising match for *neuromorphic* processors, which can efficiently update stateful neurons due to a tight integration of massively parallel compute and memory. Neuromorphic processors are an emerging class of brain-inspired hardware architectures, with notable examples like IBM's NorthPole (Modha et al., 2023), SpiNNaker 2 (Mayr et al., 2019), Tianjic (Pei et al., 2019), and Intel's Loihi 2 (Orchard et al., 2021). Beyond parallelism and compute-memory integration, different neuromorphic processors offer unique sets of further computational features, including event-driven compute and messaging, low-precision arithmetic, and support for unstructured sparse weight matrices. These sets of features offer unique opportunities to optimize and compress linear RNNs for real-world applications.

In this work, we explore the potential of unstructured sparsity–in weights and activations–and fixed-point quantization for the compression of linear RNNs and acceleration on neuromorphic hardware as illustrated in Figure 1. Specifically, we explore four key research questions:

1. Can we train linear RNNs with high synaptic and activation sparsity while retaining high performance?

2. Do highly sparse linear RNNs outperform dense linear RNNs across different inference compute budgets?

3. Can fixed-point quantization compress sparse linear RNNs without damaging the network's performance?

4. Can unstructured sparsity and fixed-point quantization be translated into latency and energy advantages on neuromorphic hardware?

We provide definite positive answers to questions 1 and 4, and present positive evidence for questions 2 and 3.

## 2. Compressing linear RNNs

### 2.1. Linear Recurrent Neural Networks

Recurrent neural networks (RNNs) are a class of neural networks designed for processing sequential data by maintaining hidden states that capture temporal dependencies. Linear RNNs distinguish themselves through their linear dynamics, which enables parallelization over the sequence length and, therefore, efficient training. Previous work has shown—both theoretically (Orvieto et al., 2024) and empirically (Gu et al., 2022a)—that the network's recurrent weight matrix can effectively be diagonalized in the complex domain without any loss of generality or model capacity. We use this diagonal formulation of linear RNNs, such that the network's update equations for the state $\mathbf{x}_k \in \mathbb{C}^N$ and output $\mathbf{y}_k \in \mathbb{R}^M$ are given by:

$$\mathbf{x}_k = \text{diag}(\bar{\mathbf{A}}) \otimes \mathbf{x}_{k-1} + \bar{\mathbf{B}}^T \mathbf{u}_k \tag{1}$$

$$\mathbf{y}_k = \bar{\mathbf{C}}^T \mathbf{x}_k + \text{diag}(\bar{\mathbf{D}}) \otimes \mathbf{u}_k \tag{2}$$

where $\otimes$ denotes the Hadamard product, $\mathbf{u}_k \in \mathbb{R}^M$ is the input sequence, $\text{diag}(\bar{\mathbf{A}}) \in \mathbb{C}^N$ are the diagonal recurrent weights, $\bar{\mathbf{B}}^T \in \mathbb{C}^{M \times N}$ are the input weights, $\bar{\mathbf{C}}^T \in \mathbb{C}^{N \times M}$ are the output weights, and $\text{diag}(\bar{\mathbf{D}}) \in \mathbb{R}^M$ are the residual weights. We follow the S5 model (Smith et al., 2023) for the initialization and parameterization of the linear RNN.

Because of its linearity, the temporal mixing of the S5 block above is followed by a nonlinear channel mixing block. We use a particular variant of the GLU block (Dauphin et al., 2017) where the linear RNN's output $\mathbf{y}_k \in \mathbb{R}^M$ is transformed as: $GLU(y_k) = \sigma\left(W\tau(\mathbf{y}_k)\right) \otimes \tau(\mathbf{y}_k)$ where $\tau$ is an element-wise nonlinear function (we use either the Gaussian error linear unit (GELU) or the Rectified Linear Unit (ReLU)), $W \in \mathbb{R}^{M \times M}$ is a weight matrix, and $\sigma$ is the sigmoid function. The full model architecture is illustrated in Figure 2.

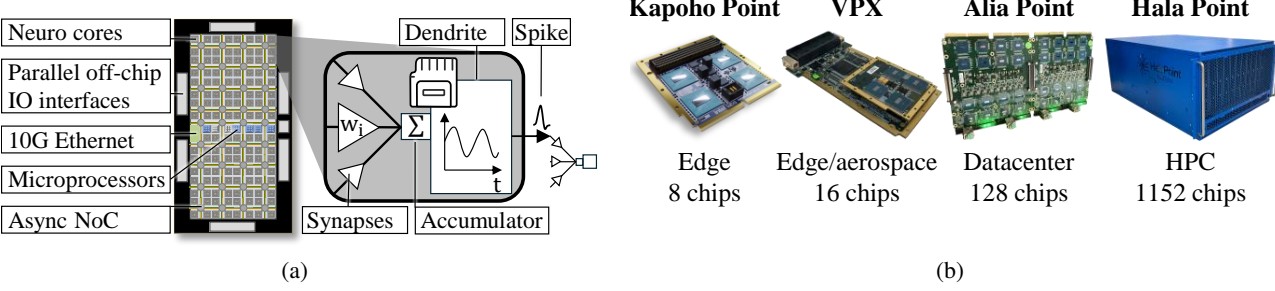

*Figure 3.* (a) Loihi 2 implements a network of neurons, which are processed by neuro-cores and communicate via an asynchronous network-on-chip. Parallel IO and $10\,\mathrm{Gbit}$ Ethernet interfaces enable a Loihi 2 chip to communicate with other Loihi 2 chips and external hosts, respectively. Embedded microprocessors provide a flexible method of interaction with neuro-core registers, management, and communication. On a neuro-core, each neuron receives spike messages from other neurons via synapses with multiplicative weights $w_i$, and sums them up by one or multiple dendritic accumulators. The input is used by a dendrite to update memory states that are local to the respective neuron. The neuron communicates with other neurons by sending spike messages. (b) Different Loihi 2 systems are available to cover a wide range of applications from the edge to HPC with up to $1\,\mathrm{B}$ neurons.

## 2.2. Neuromorphic Computing with Intel Loihi 2

Neuromorphic processors mimic computing principles of the brain, which excels in processing sequential data streams with just around $20\,\mathrm{W}$ of power. Loihi 2 is the second-generation of Intel's neuromorphic research processor (Orchard et al., 2021) and implements a spiking neural network as illustrated in Figure 3. The network is processed by massively parallel compute units, with 120 *neuro-cores* per chip. The neuro-cores compute and communicate asynchronously, but a global algorithmic time step is maintained through a barrier synchronization process. The neuro-cores are co-located with memory and can thus efficiently update local states, simulating up to $8\,192$ stateful neurons per core. Each neuron can be programmed by the user to realize a variety of temporal dynamics through assembly code. Input from and output to external hosts and sensors is provided with up to $26\,\mathrm{M}$ 32 bit integer $\mathrm{messages/s}$ (Shrestha et al., 2024b). Loihi 2 can scale to real-world workloads of various sizes with up to $1\,\mathrm{B}$ neurons and $128\,\mathrm{B}$ synapses, using fully-digital stacked systems shown in Figure 3.

The architectural features of Loihi 2 offer unique opportunities to compress and optimize deep learning models. Like GPUs, its neuro-cores benefit from model quantization, as it supports low-precision arithmetics, $8\,\mathrm{bit}$ for synaptic weights and up to $32\,\mathrm{bit}$ for spike messages. Unlike GPUs, Loihi 2 is optimized for local computations within neurons, a common focus of neuromorphic processors. First, it allows fast and efficient updates of neuronal states with recurrent dynamics with minimal data movement, due to its tight compute-memory integration. Second, the fully asynchronous event-driven architecture of Loihi 2 allows it to efficiently process unstructured sparse weight matrices. Third, the neuro cores can leverage sparsified activation between neurons, as the asynchronous communication transfers only non-zero messages.

## 2.3. Evaluating Benefits from Sparsity

Unstructured sparsity has demonstrated compelling results as an effective model compression technique, serving both as a framework for theoretical analysis of sparsity algorithms and as an upper-bound for the gains achievable with constrained forms of sparsity (Liu & Wang, 2023; Mishra et al., 2021; Han et al., 2015). In particular, when compared to structured sparsity patterns, like N:M (Mishra et al., 2021) or block-diagonal, it typically attains higher task performance or compression rates (Lee et al., 2023). However, the gains of unstructured sparsity have not been realized as the traditional GPU architecture is suited to exploit only block sparsity structures (Liu & Wang, 2023). Additionally, sparse activations complement synaptic sparsity, resulting in fewer operations overall (Mukherji et al., 2024), but GPUs typically cannot take advantage of activation sparsity either. Realizing the benefits of unstructured sparsity requires suitable hardware architectures (Lie, 2023; Ashby et al., 2019; Zhang et al., 2021). The event-driven neuromorphic architecture of Loihi 2 is inherently suited to take advantage of the unstructured sparsity in both connections and activities, in particular, when they are extremely sparse, *i.e.,* $\geq 90\%$. Therefore, we choose to compare the benefits of efficiency gained from sparsity on Loihi 2 with equivalent dense networks on an edge GPU.

Theoretical studies have shown that wider sparse layers outperform dense layers with the same number of parameters (Golubeva et al., 2021; Chang et al., 2021). Research has further shown that, in practice, it is better to train a larger over-parameterized network and prune it to make it leaner compared to training a compact sparse network from start (Frankle & Carbin, 2018; Renda et al., 2020; Chen et al., 2020). There is evidence showing minimal loss in accuracy when the networks are pruned, typically to sparsity levels of 50–80% (Chen et al., 2020). However, there is not much

research on performance at extreme levels of sparsity of $\geq 90\%$. We thus ask; *Do highly sparse networks achieve superior performance to dense networks when operating under identical inference compute budgets? How does the performance benefit of sparsity vary with increased compute budget?*

In Section 3.2, we evaluate the effect of pruning and activity sparsification on multiply-and-accumulate (MACs) operations and task performance for a $k$-family of sparse and densely trained networks where $k_{\text{sparse}} \in [0.5, 3.0]$, $k_{\text{dense}} \in [0.25, 1.0]$ is the width scaling factor of the networks. In linear layers, which account for most of the computation in the S5 architecture, MACs scale linearly with weight and pre-activation sparsity. The detailed MAC calculation is reported in Appendix A.1. Additionally, we benchmark iso-accuracy models on relevant hardware to validate the theoretical gains from sparsity with latency and power measurements in Section 3.3.

### 2.4. Model Compression

**Synaptic pruning**  Given our focus on edge and low-latency applications, we design our compression pipeline assuming that fine-tuning or re-training of the models is feasible. Following previous work (Mishra et al., 2021), we initialize the parameters from the pre-trained dense models. We adopt iterative magnitude pruning (IMP) which increases sparsity progressively during training and achieves better task performance than one-shot approaches, especially at high sparsity levels (Zhu & Gupta, 2018; Lee et al., 2023).

Specifically, we train for $E$ epochs with $T$ update steps in total. Sparsity starts at $S_i = 0$ at $t_i = 0$ and is increased following a degree-3 polynomial schedule (Zhu & Gupta, 2018) and updated three times per epoch as:

$$S_t = S_f - (S_f - S_i) \cdot \left(1 - \frac{t - t_i}{t_f - t_i}\right)^3$$

with $t_f = 0.75T$. Given the total sparsity $S_t$ and weights $W_t^\ell \in \mathbb{R}^{N^\ell \times M^\ell}$ at time $t$ and position $\ell$ in the network, we scale the sparsity $s_t^\ell$ for each weight according to the Erdös-Renyi-Kernel (ERK) strategy (Evci et al., 2020; Mocanu et al., 2018) to compute the mask $M_t^\ell$:

$$s_t^\ell = s_t \cdot \frac{N^\ell + M^\ell}{N^\ell \cdot M^\ell}$$
$$M_t^\ell = \mathbb{1}\left(|W_t^\ell| \geq \tau_t^\ell\right)$$
$$\tau_t^\ell = \min\left[\text{TopK}\left(|W_t^\ell|, s_t^\ell N^\ell M^\ell\right)\right]$$

where $\tau_t^\ell$ is the calculated threshold for $W_t^\ell$ to reach sparsity $s_t^\ell$ and $\text{TopK}(W, k)$ gives the top-$k$ values from $W$. In the forward pass, weights are masked as $\bar{W} = M \odot W$, while the backward pass applies straight-through estimation (Bengio et al., 2013), enabling gradient updates also for masked weights.

**Activity sparsification**  Sparsifying layer activations provide another means for reducing the compute and on-chip memory requirements during inference. In particular, sparse pre-activations of linear layers can significantly reduce the number of MACs required for the associated matrix-vector multiplication (MVM), if appropriately supported by the hardware backend. On sparse and event-driven accelerators, such as Loihi 2, sparse pre-activations directly translate into MACs savings since the MVM operation is computed as

$$MVM(W, x) = W_{\{i,j|x_j \neq 0\}} x_{\{i|x_i \neq 0\}} \tag{3}$$

In contrast, GPU architectures struggle to leverage dynamic sparse activation patterns and have demonstrated gains with more structured activation patterns, and only in memory-bound regimes as in auto-regressive generation with large models (Mirzadeh et al., 2024; Zhang et al., 2024; Shazeer et al., 2017; He, 2024a).

Techniques for activation sparsity include top-k (Key et al., 2024), sigma-delta coding (Shrestha et al., 2024a; O'Connor & Welling, 2016), sparse mixture-of-experts (Fedus et al., 2022; He, 2024b) and *ReLU-fication* (Mirzadeh et al., 2024). We base our methodology on the latter of these. Since ReLU is a fully element-wise operation, it doesn't require synchronization across channels which would complicate implementation in compute-memory integrated platforms, such as Loihi 2. Following previous work on transformer models (Mirzadeh et al., 2024), we start from the original dense model with GELU non-linearity, as shown in Figure 2, and apply two modifications. First, we replace the GELU activation with a ReLU, sparsifying pre-activations of the linear layer in the GLU block. Second, we insert additional ReLU activations after the residual add in the GLU block and to the real component of the S5 hidden layer, further increasing the pre-activation sparsity of linear operators. Both model surgeries are applied to the pre-trained model at the beginning of the iterative pruning procedure, enabling accuracy recovery from both weight and activation pruning without extra training budget.

**Quantization and fixed-point computation**  Reducing the numerical precision of weights and activations through quantization is an essential way to compress machine learning models, directly leading to reduced memory footprint and faster inference (Gholami et al., 2021). We denote the tensor to be quantized with $\mathbf{x}$ and the number of bits to use with $n$, such that the quantized tensor $\bar{\mathbf{x}}_n$ is defined as:

$$\bar{\mathbf{x}}_n = \left\lfloor \frac{\mathbf{x}}{\Delta_x} + z_x \right\rceil = \lfloor s_x \mathbf{x} + z_x \rceil \tag{4}$$

where $\lfloor \cdot \rceil$ indicates rounding to the nearest integer, $s_x$ is the scale for the given tensor, $z_x$ is the zero point, and $\Delta_x$ is the corresponding step size. For simplicity, we choose

$s_x = (2^{n-1} - 1)(\max |\mathbf{x}|)^{-1}$ and $z_x = \mathbf{0}$, *i.e.*, we use symmetric quantization based on the absolute maximum.

Post-training quantization (PTQ) applies quantization to a pre-trained model without further training, which is computationally efficient but may lead to a notable drop in accuracy, especially for complex models or tasks (Gholami et al., 2021). Without constraints during training, it has been shown to under-perform on both nonlinear (Wu et al., 2016) and linear RNNs (Abreu et al., 2024). In contrast, quantization-aware training (QAT) incorporates quantization into the training process using straight-through estimators for the gradients (Bengio et al., 2013), allowing the model to adapt to the reduced precision and typically achieving superior performance retention compared to PTQ (Hubara et al., 2018), which has also shown promising results on linear RNNs such as S4D (Meyer et al., 2024) and S5 (Abreu et al., 2024) on synthetic tasks from the Long Range Arena benchmark (Tay et al., 2021). To demonstrate advantages on hardware, we use static quantization (Gholami et al., 2021) using only fixed-point (integer) arithmetic (Wu et al., 2020). Whereas in dynamic quantization, scales $s_x$ are computed dynamically on incoming data (and therefore requiring floating-point operations), static quantization pre-computes scales for all weights and activations in the neural network and "freezes" these scales so that the network can be converted to use only fixed-point arithmetic.

Following prior work on quantizing linear RNNs (Abreu et al., 2024), we choose $8\,\mathrm{bit}$ for all weights, except the diagonal recurrent $\mathrm{diag}(\bar{A})$ weights which is stored with $16\,\mathrm{bit}$. All activations are quantized to $16\,\mathrm{bit}$. We denote this quantization recipe with W8A16. This is a more compressed quantization scheme than previous work that deployed a linear RNN to fixed-point hardware using W8A24 (Meyer et al., 2024). For the linear RNNs that are deployed to the Loihi 2 chip, we combine QAT with sparse training.

### 2.5. Porting S5 to Loihi 2

Running S5 on Loihi 2 requires a range of adjustments, to fully leverage the neuromorphic architecture and to adhere to its constraints. As a result, the S5 network shown in Figure 2 is transformed into a network of synapses and neurons for Loihi 2 as illustrated in Figure 8. In general, a state vector of dimension $\mathbb{R}^M$ is encoded by M neurons. Matrix-vector multiplications are hardware accelerated by the synaptic layers, which take a vector of neuron activities, multiply it with the matrix of synaptic weights, and pass the output to the next layer of neurons. Since complex numbers are not natively supported on Loihi 2, the complex matrices $\bar{B}$ and $\bar{C}$ have been split into two synaptic layers each. Similarly, the complex state $x_k$ is stored by two neuronal states. The remaining operations are performed within the assembly-programmable neurons.

A single layer of programmable neurons can efficiently fuse many operations on the vector it encodes. This applies to all element-wise operations where each neuron must operate only on its local states. The neuronal layers thus implement ReLUs, BatchNorm, Hadamard products, residual add, and multiplications of a state vector with a diagonal matrix. Applying this layer fusion, the full S5 architecture only requires one neuron group for the encoder, one for the decoder, and three for each S5 block. The detailed mapping of operations to neuron groups is illustrated in Figure 8,

## 3. Results

### 3.1. Experimental Setup

**Software** We implemented our methodology in JAX 0.4.30, building on top of the original S5 codebase (Smith et al., 2023), with JaxPruner (Lee et al., 2023) for the pruning algorithms and the AQT library (Google, 2024) for quantization-aware training. We implemented static quantization and a fixed-point model ourselves using only JAX.

**Audio denoising task** We evaluated our approach on the Intel Neuromorphic Deep Noise Suppression Challenge (Timcheck et al., 2023). The objective of the Intel N-DNS Challenge is to enhance the clarity of human speech recorded on a single microphone in a noisy environment. The Intel N-DNS Challenge utilizes data from the Microsoft DNS Challenge, encompassing clean human speech audio samples and noise source samples. (Reddy et al., 2020; 2021a;b; Dubey et al., 2024). Clean human speech and noise samples are mixed to produce noisy human speech with a ground truth clean human speech goal.

To train our models, we used the default Intel N-DNS Challenge training and validation sets, each consisting of $60\,000$ noisy audio samples of $30\,\mathrm{s}$ each, and a test set with $12\,000$ samples. We encoded and decoded each audio sample using the Short-Time Fourier Transform (STFT) and Inverse Short-Time Fourier Transform (iSTFT) (Gröchenig, 2013). Following the N-DNS baseline solution, NsSDNet (Shrestha et al., 2024a), we adopted a $32\,\mathrm{ms}$ window length and a $8\,\mathrm{ms}$ hop length for the STFT/ISTFT. This resulted in a nominal real-time audio processing latency of $32\,\mathrm{ms}$, which allows ample time ($8\,\mathrm{ms}$) for denosing network inference, as $40\,\mathrm{ms}$ is the standard for an acceptable latency according to the Microsoft DNS Challenge. We evaluated the denoising quality of our model using the scale-invariant signal-to-noise ratio (SI-SNR)

$$\text{SI-SNR} = 10 \log_{10} \frac{\|s_{\text{target}}\|^2}{\|e_{\text{noise}}\|^2}. \tag{5}$$

Importantly, SI-SNR provides a volume-agnostic measure of audio cleanliness relative to the ground truth signal.

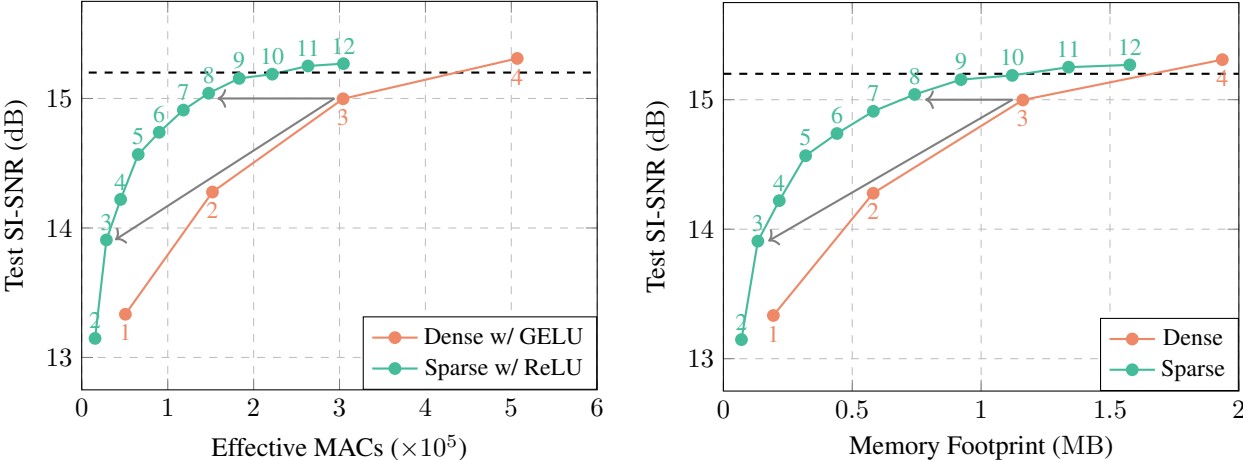

*Figure 4.* Pareto fronts for S5 network audio denoising quality (SI-SNR) as a function of effective compute (left) and memory footprint (right) on the Intel N-DNS test set. S5 networks with weight and activation sparsity (green) exhibit a large domain of Pareto optimality versus dense S5 networks (orange). Number annotations enumerate increasing S5 dimensionality configurations, from $500\,\mathrm{k}$ to $4\,\mathrm{M}$ parameters. Dashed horizontal like marks SI-SNR of Spiking-FullSubNet XL, the previous state-of-the-art model. The horizontal arrows highlight models used for hardware deployment, the diagonal arrows highlight models of the same width. See text for details.

### 3.2. Pareto Front of Performance and Efficiency

We studied the performance-efficiency Pareto front of dense and sparse models across inference compute budgets. Starting from the S5 architecture (Smith et al., 2023), we trained a family of dense models of increasing size by linearly scaling the model dimensions (i.e. model width and size of the SSM hidden state), while keeping the depth fixed to three S5 layers. Similarly, we trained a family of sparse models, i.e., pruned and ReLU-fied, according to our methodology discussed above, with $90\%$ of weights pruned by the end of training (further details on the model dimensions are provided in Appendix A.2). The results, reported in Figure 4, compare de-noising performance (SI-SNR) and computational efficiency as measured by effective MACs and memory footprint (see Appendix A.1).

The results show that sparsification significantly degrades performance when applied to under-parametrized dense models (e.g., sparsifying dense-3 reduces SI-SNR by $7.3\%$). However, task performance is recovered with increased model dimensions and the accuracy of dense models is matched by larger sparse ones, with fewer MACs and lower memory requirements. This gives empirical support to theoretical work on the capacity of sparse-and-wide neural networks (Golubeva et al., 2021). For example, sparse-8 model requires $\mathbf{2\times}$ **less compute** and $\mathbf{36\%}$ **lower memory** than the dense-3 model, **while achieving the same level of accuracy**. Overall, sparse models constitute the Pareto front of task performance and computational efficiency across compute budgets.

In terms of absolute task performance, we find that the S5 architecture provides state-of-the-art results on audio denois-

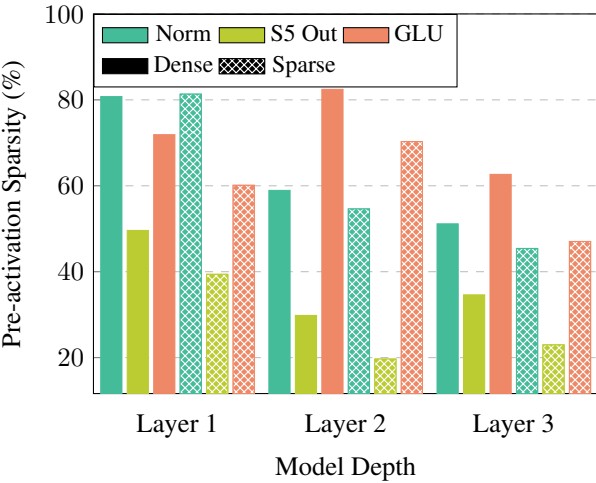

*Figure 5.* Activation sparsity of ReLU blocks across model depth for a dense model and a sparse-weight model. The sparse-weight model exhibits significantly lower activation sparsity across layers.

ing out of the box. When compared to Spiking-FullSubNet-XL (Hao et al., 2024), the Track 1 winner of the Intel N-DNS Challenge with $15.2\,\mathrm{dB}$ SI-SNR, our sparse-11 S5 model requires $\mathbf{3.2\times}$ **less compute** and $\mathbf{5.37\times}$ **lower memory iso-accuracy**. This finding is in line with previous research on audio modeling with state space models (Goel et al., 2022), and provides additional evidence on the suitability of these architectures for signal processing.

**Interaction of weight and activation sparsity** An interesting question is what is the interaction between the two

types of sparsity, in weights and activations. Figure 5 reports the pre-activation sparsity for different layers across the model depth for two ReLU-fied models of the same size (model variant 6), with and without synaptic sparsity. We observe that the synaptic-sparse model exhibits lower activation sparsity across the board, a finding that is consistent across model sizes. In addition, activation sparsity significantly decreases with model depth, both for dense and sparse models. These phenomena, previously observed in other models (Mukherji et al., 2024), suggest that, during training, the model compensates the reduced information flow caused by pruning with increased levels of activation. Additional research on more advanced activation functions would allow for the optimal allocation of MACs, especially those that provide explicit control over sparsity without cross-channel synchronization (e.g., approximate top-k (Key et al., 2024)).

### 3.3. Hardware Implementation

**Impact of fixed-point conversion**   Since Loihi 2 only supports fixed-point (FXP) arithmetic, as presented in Section 2, we quantized the weights and activations of our model and implemented the network dynamics in FXP arithmetic. The effect of our quantization methodology is presented in Figure 6. Starting from a 32-bit floating-point (FP32) model, we apply static quantization, which rounds weights and activations using fixed scales, but still performs the actual computation in FP32. Notably, Quantization-Aware Training (QAT) is very effective in maintaining test performance (SI-SNR) from FP32 to static quantization, compared to Post-Training Quantization (PTQ). The frozen scales from static quantization are imported into our FXP model implemented in JAX, which uses only int32 types and fixed-point arithmetic to compute the forward pass of the model. We observe further performance degradation in the FXP simulation, which we analyze in more detail in Appendix A.3.3. We finally map the FXP model to Loihi 2 and perform inference on the chip, again finding a degradation in SI-SNR, which is likely due to subtle differences in the integer arithmetic performed by the FXP simulation and Loihi 2 implementation with fused layers. Another source of mismatch is that the FXP model in simulation handles overflows by clipping to the maximum value, whereas Loihi 2 "wraps around" the value, resulting in a sign inversion. The size of the model decreases by about a factor of 4 when transitioning from FP32 weights to INT8 weights, as shown on the right side of Figure 6.

**Power and Performance**   To measure the empirical efficiency benefits afforded by the sparse S5 model on neuromorphic hardware, we profile inference on Loihi 2 using the fixed-point S5 model, in particular, configuration sparse-8 from Figure 4. To compare to conventional hardware, we profile the smallest dense model that achieves equivalent

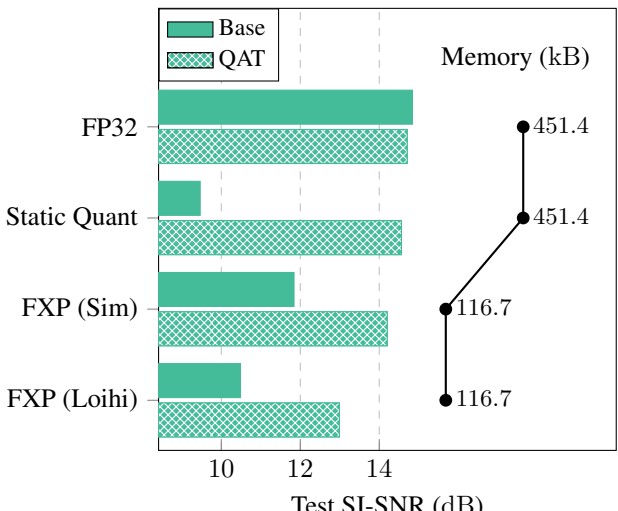

*Figure 6.* Impact of quantization interventions on Test SI-SNR and memory footprint, with and without quantization-aware training, for model variant sparse-6. The results show that the Base model without QAT performs slightly better in FP32 than the QAT model, but significantly worse in static quantization and fixed-point precision.

performance on Jetson Orin Nano[1], which is configuration dense-3 from Figure 4. There exist a variety of modes in which to execute a model on Loihi and Jetson, each exhibiting different tradeoffs in terms of latency, throughput, and energy. Therefore, we present different modes for a comprehensive characterization and comparison. We summarize our profiling results in Table 1. More details on the different execution modes on Loihi 2 are presented in Appendix A.3.2.

In real-time, token-by-token processing on a single input sequence, Loihi 2 processes a single STFT frame **35× faster** and with **1200× less energy** than the Jetson Orin Nano. When the Jetson Orin Nano processes "chunks" of multiple time steps, its utilization increases, and energy per token improves. With the largest chunks that fit the real-time requirement of latency $\leq 8\,\mathrm{m\,sec}$, Loihi 2 is **42× faster** and uses **149× less energy** per token.

In offline processing, when many STFT frames are buffered to process in succession (or in parallel), the energy efficiency and throughput of the Jetson Orin Nano improves. Loihi 2 performs offline processing with pipelining (see Appendix A.3.2 for further explanation). When processing single sequences, *i.e.* batch size $b = 1$, Loihi 2 has **3.7× higher throughput** with **8× less energy** per sample.

It is important to note that the Jetson Orin Nano is only fully

---

[1]Our W8A16 fixed-point model in JAX does not provide a speedup over the FP32 model on the Jetson Orin Nano, therefore we profile the FP32 model.

*Table 1.* Power and performance results[*]. The Loihi 2 is running a sparse and quantized S5 model, while the Jetson Orin Nano is running a smaller dense S5 model that reaches similar test performance. All measurements are averaged over 8 random samples from the test set, each containing 3 750 time steps. Gray highlights denote violation of real-time constraints for the audio denoising task. Best real-time results are underlined.

| | Mode | Latency ($\downarrow$) | Energy ($\downarrow$) | Throughput ($\uparrow$) |
|---|---|---|---|---|
| **Token-by-token** | | | | |
| Intel Loihi 2[†] | Fall-Through | 76 µs | 13 µJ/tok | 13 178 tok/s |
| Jetson Orin Nano[‡] | Recurrent 1-step ($b = 1$) | 2 688 µs | 15 724 µJ/tok | 372 tok/s |
| Jetson Orin Nano[‡] | Recurrent 10-step ($b = 1$) | 3 224 µs | 1 936 µJ/tok | 3 103 tok/s |
| Jetson Orin Nano[‡] | Recurrent 100-step ($b = 1$) | 10 653 µs | 626 µJ/tok | 9 516 tok/s |
| Jetson Orin Nano[‡] | Recurrent scan ($b = 1$) | 236 717 µs | 404 µJ/tok | 15 845 tok/s |
| **Sample-by-sample** | | | | |
| Intel Loihi 2[†] | Pipeline | 60.58 ms | 185.80 mJ/sam | 16.58 sam/s |
| Jetson Orin Nano[‡] | Scan ($b = 1$) | 233.48 ms | 1 512.60 mJ/sam | 4.28 sam/s |
| Jetson Orin Nano[‡] | Scan ($b = b_{max}$) | 226.53 ms | 5.89 mJ/sam | 1 130.09 sam/s |

[†] Loihi 2 workloads were characterized on an Oheo Gulch system with N3C1-revision Loihi 2 chips running NxCore 2.5.8 and NxKernel 0.2.0 with on-chip IO unthrottled sequencing of inputs. Researchers interested to run S5 on Loihi 2 can gain access to the software and systems by joining *Intel's Neuromorphic Research Community*. [‡] Jetson workloads were characterized on an NVIDIA Jetson Orin Nano 8GB running Jetpack 6.2, CUDA 12.4, JAX 0.4.32, using the MAXN SUPER power mode; energy values are computed based on the TOT power as reported by jtop 4.3.0. The batch size $b_{max} = 256$ was chosen to be the largest that fits into memory. [*] Performance results are based on testing as of January 2025 and may not reflect all publicly available security updates; results may vary.

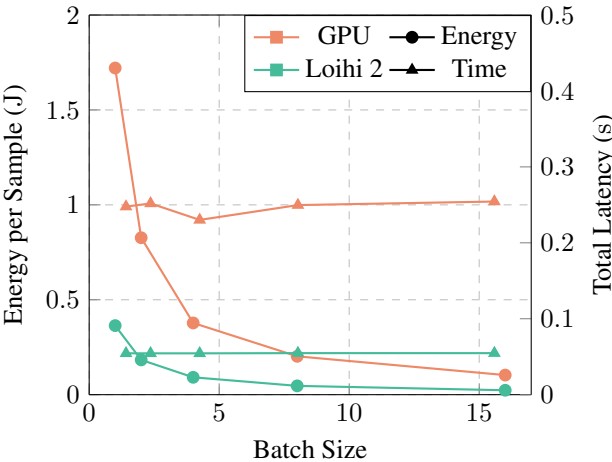

*Figure 7.* Impact of batching on energy efficiency and latency for Loihi 2 and Jetson Orin Nano. Both platforms exhibit similar trends: energy per sample decreases proportionally with batch size, while latency remains approximately constant. Loihi 2 maintains competitive performance on both metrics across batch sizes.

utilized when processing 256 sequences in parallel, and at this level, it shows significantly higher throughput while consuming less energy per sample, compared to Loihi 2. We include these results in the last row of Table 1.

**Impact of batch processing** While several edge applications typically require batch-one inference, some scenarios can benefit from support for small-batch processing, e.g., de-noising audio streams from multiple on-device micro-

phones. For this reason, it is interesting to investigate the effect of batch processing on energy efficiency and latency for the two hardware architectures. Intel Loihi 2 doesn't natively support batching in the sense of processing multiple independent samples through the same model instantiation. However, the parallel inference of independent sequences can be achieved by replicating the model on the chip as many times as required by batch size, thereby obtaining higher throughput through a larger silicon area. We extended the results in Table 1 to compare the effect of this implementation of batching on a 16-chip Loihi 2 VPX board to the usual batch processing of the Jetson Orin GPU. The results, reported in Figure 7, show the energy per sample and the total latency for both architectures across batch sizes, from 1 to 16. Both hardware backends exhibit a similar trend: while total latency remains constant, the energy efficiency improves proportionally with batch size. Loihi 2 remains competitive across batch sizes, showing between 4.43 to 4.72× lower energy per sample and 4.52× lower latency on average. It is important to note that since model replicas are physically mapped to different cores on Loihi, the resource requirements increase linearly with batch size. For this reason, such batch processing on Loihi is only feasible for small models and small batch sizes.

**Energy at real-time inference rate** The latency budget for the neural network component of the audio denoising pipeline, running either on Loihi 2 or on the Jetson, is 8 ms. Our Loihi 2 and Jetson implementations are well below 8ms for online inference. Thus, to estimate the energy consumption in real-time settings, where subsequent tokens

are actually $8\,\text{ms}$ apart, we rescale the power as:

$$P_{\text{total}}^{\text{real-time}} = P_{\text{static}} + \frac{t_{\text{compute}}}{8\,\text{ms}} P_{\text{dynamic}},$$

based on the power measurements in token-by-token processing. In this setting, Loihi 2 achieves $1\,128\,\mu\text{J}/\text{tok}$ while the Jetson achieves $36\,528\,\mu\text{J}/\text{tok}$ for token-by-token processing and $3\,720\,\mu\text{J}/\text{tok}$ when processing chunks of 10 time steps at once. Loihi 2 remains at least $3\times$ more energy efficient than the Jetson Orin Nano.

**Limitations**  Our Jetson Orin Nano implementation is in FP32, while our Loihi 2 implementation is in W8A16. Our fixed-point model in JAX provides no improvements in runtime or energy. More competitive Jetson energy, latency, and throughput could potentially be obtained by developing a more optimized quantized implementation.

## 4. Discussion

In this work, we explored the Pareto front of efficiency and performance for a streaming audio processing task, comparing dense and sparsified variants of a linear RNN based on the S5 architecture. We showed that combining activation sparsity and unstructured weight pruning results in a significant reduction in compute requirements, up to $3.2\times$, and memory footprint, $5.7\times$, without accuracy degradation. In addition, we validated these theoretical gains with a hardware-accelerated implementation on a compute-memory integrated coarse accelerator, the Intel Loihi 2 neuromorphic chip. When quantized and deployed on Loihi 2, sparse models deliver $42\times$ lower latency and $149\times$ lower energy consumption in token-by-token processing, compared to the iso-accuracy dense models on the Jetson Orin Nano GPU.

In conclusion, our work demonstrates that sparse event-driven accelerators, such as neuromorphic processors, can provide state-of-the-art accuracy on high-frequency signal processing tasks, with orders of magnitude gains in latency and energy efficiency. This possibility opens up several research directions to further materialize these gains in real-world applications. In particular, future work should investigate how the efficiency-performance Pareto front scales up to larger models and more complex tasks, such as language and multimodal modeling. In this setting, the scalability of multi-chip neuromorphic processors (Kudithipudi et al., 2025) and high-frequency execution could power the growing need for large-scale inference compute (Snell et al., 2024). Finally, improvements to our fixed-point conversion methodology and the use of advanced data types (e.g. FP8), could help close the gap between simulation and hardware deployment.

## Acknowledgments

We acknowledge the support of several colleagues at Intel who enabled our experimental setup on Intel Loihi 2, both in hardware and software. We are also grateful to Prof. Eyke Hüllermeier for feedback on an early draft of this work.

## Impact Statement

This paper presents work whose goal is to advance the field of Machine Learning. There are many potential societal consequences of our work, none which we feel must be specifically highlighted here.

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

# A. Supplemental Material

## A.1. Effective MACs computation for S5 architecture

In this section, we detail the computation of effective multiply-accumulate operations (MACs) for different components of the S5 architecture. The total MAC count provides an estimate of the computational cost associated with each stage of the model. Below, we outline the individual contributions from key components of the architecture. The effective MACs for all model sizes–sparse and dense–in Figure Figure 4 are calculated based on the formulas below, summed over the entire network structure.

**Notation:**

- $N_{\text{input}}$: Input dimension

- $N_{\text{model}}$: Model dimension for activations outside of the linear RNN.

- $N_{\text{ssm}}$: Dimension of the linear RNN's hidden state.

- $N_{\text{output}}$: Output dimension (equal to the number of classes for classification)

- $d_x^{\text{wgt}}$: Density of weights for $x$

- $d_x^{\text{act}}$: Density of activations for $x$

where the density $d$ is calculated from the sparsity $s$ as $d = 1 - s$.

**Breakdown of MAC Calculation per Component:**

- **Encoder:** The MACs for the encoder depend on the input dimension, model size, and scale linearly with activation and weight densities:5

$$N_{\text{input}} N_{\text{model}} d_{\text{encoder}}^{\text{wgt}} d_{\text{input}}^{\text{act}} \tag{6}$$

- **Batch Normalization (BatchNorm):** A lightweight operation, requiring only element-wise scaling, leading to:

$$N_{\text{model}} \tag{7}$$

- **S5 Hidden Layer:** The hidden state update for the S5 model involves both matrix multiplications and element-wise operations:

$$2 N_{\text{model}} N_{\text{ssm}} d_B^{\text{wgt}} d_{\text{pre\_ssm}}^{\text{act}} + 4 N_{\text{ssm}} \tag{8}$$

- **SSM Output Layer:** Computes the output transformation of the linear RNN:

$$2 N_{\text{ssm}} N_{\text{model}} d_C^{\text{wgt}} d_{\text{hidden}}^{\text{act}} + N_{\text{model}} d_{\text{pre\_ssm}}^{\text{act}} \tag{9}$$

- **Gated Linear Unit (GLU):** The computation for the GLU involves matrix multiplications for the dense weight matrix, followed by an element-wise multiplication:

$$N_{\text{model}}^2 d_{\text{GLU}}^{\text{wgt}} d_{\text{pre\_GLU}}^{\text{act}} + N_{\text{model}} \tag{10}$$

- **Classification Head:** The final linear projection for classification:

$$N_{\text{model}} N_{\text{output}} d_{\text{head}}^{\text{wgt}} d_{\text{pre\_hread}}^{\text{act}} \tag{11}$$

- **Regression Head:** The regression head follows the same computation as the classification head:

$$N_{\text{model}} N_{\text{output}} d_{\text{head}}^{\text{wgt}} d_{\text{pre\_hread}}^{\text{act}} \tag{12}$$

Numerical operations such as the inverse square-root, sigmoid function, and others, are ignored from our MAC calculations, as is commonly done when calculating the MACs or floating point operations (FLOPs) of machine learning models (Evci et al., 2020).

## A.2. Experimental Details

**Model architecture** Our linear RNN is based on the S5 architecture (Smith et al., 2023), as described in Section Section 2.1. We use the following dimensions for our base model with width scaling $k = 1$ (*i.e.* configuration 4 in Figure 4). We use three layers, the recurrent state vector is $\mathbf{x}_t \in \mathbb{R}^{256}$, we use a model dimension of 192. Both input and output have dimension 257. The width scaling factors $k_i$ scale the model and recurrent state dimension linearly. In Figure 4, we report results for a $k$-family of sparse and densely trained networks where $k_{\text{sparse}} \in [0.5, 3.0]$, $k_{\text{dense}} \in [0.25, 1.0]$.

**Training recipe** We trained all models for 50 epochs with the Adam optimizer. The parameters of the SSM block were updated with initial learning rate 0.002, while the rest of the architecture used initial learning rate 0.008 and weight decay 0.04. All learning rates used cosine annealing and no warmup epochs. The dropout was set to 0.1.

## A.3. Additional Results

### A.3.1. KEYWORD SPOTTING

We extended our experiments by applying the proposed scaling protocol to the keyword spotting task of the Speech-Commands V2-35 dataset (Warden, 2018). The results, reported in Figure 9, exhibit a similar trend to that observed on the N-DNS dataset. Sparse models are more efficient while reaching the same level of accuracy. However, further scaling of the sparse model family would be required to compare against dense models at higher accuracy.

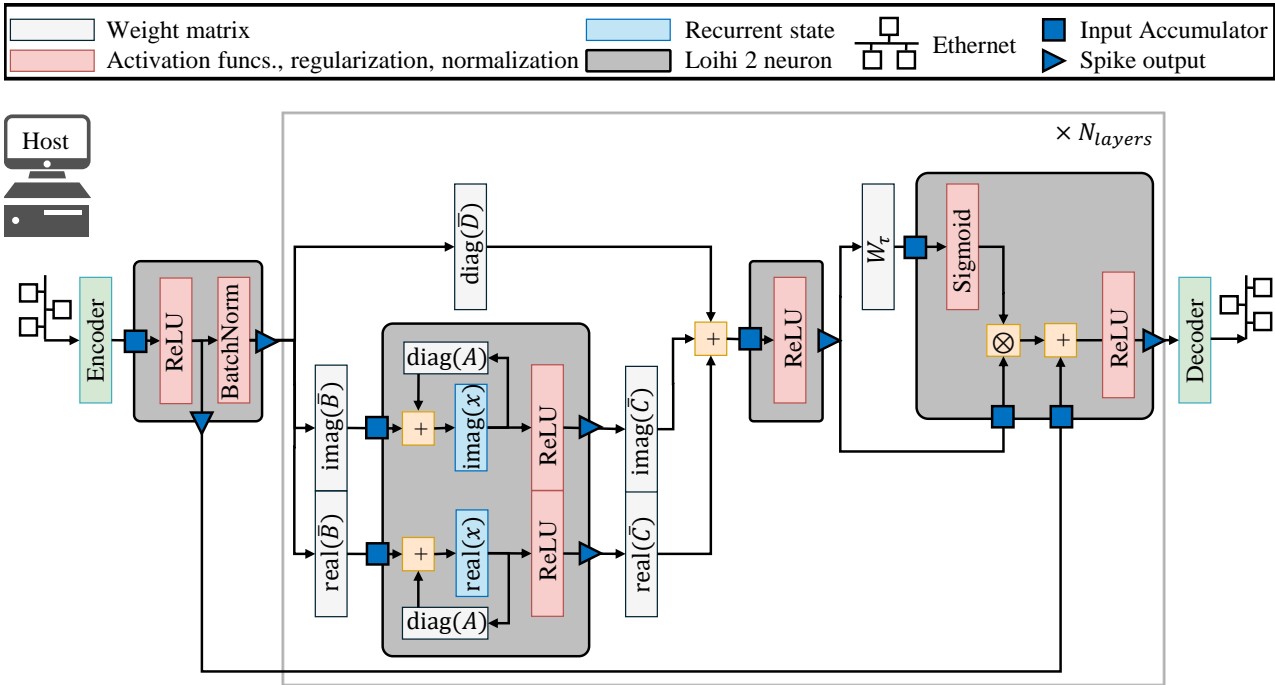

*Figure 8.* Diagram of S5 as implemented on Loihi 2. To leverage the neuromorphic hardware architecture, several adjustments are made in comparison to the original S5 model shown in Figure 2: First, complex numbers are split into real and complex components for processing. Second, ReLUs are introduced to increase activation sparsity. Third, multiple element-wise operations are fused into single neuromorphic neurons. Symbols are shown as defined in Section 2.1.

### A.3.2. LOIHI EXECUTION MODE

Loihi 2's asynchronous architecture allows to trade off between throughput and latency, as illustrated in Figure 10a. For optimal throughput, new input is provided every time step and forwarded through the neuronal layers in a pipelined mode. For optimal latency, new input is injected only once the previous input has been processed by, or fallen through, the network as fast as possible. The pipelined and fall-through mode can be balanced by changing the rate of new input, to match the throughput of a given input stream while minimizing its processing latency.

As audio denoising is typically deployed in realtime in an online fashion where one STFT input frame in processed at a time, fall-through mode is appropriate, as one desires a corresponding output STFT frame immediately.

We see that Loihi 2 processes a single STFT frame 35× faster and with 1200× less energy than the Jetson Orin Nano (Token-by-token; Loihi 2 Fall-Through and Jetson Orin Nano Recurrent 1-step (b=1) in Table 1).

### A.3.3. FIXED-POINT MODEL MISMATCH

The mismatch in Figure 6 indicates that fixed-point implementation in JAX does not perfectly match the original FP32 model when using the scales computed through our static quantization step. Further investigations show that the mismatch between hidden activations is highest for the hidden states $\mathbf{x}_k$ of the linear RNN and its outputs $\mathbf{y}_k$, see Figure 11. This mismatch increases approximately linearly with model depth, indicating that quantization errors accumulate as information propagates through the network layers. This linear escalation of errors underscores a critical challenge in fixed-point quantization of recurrent models (Wu et al., 2016; Abreu et al., 2024; Li & Alvarez, 2021; Pierro & Abreu, 2024). Consequently, ensuring the fidelity of deeper Linear RNNs on fixed-point neuromorphic hardware may require advanced quantization techniques or error mitigation strategies to preserve the network's temporal dynamics and memory capacity effectively.

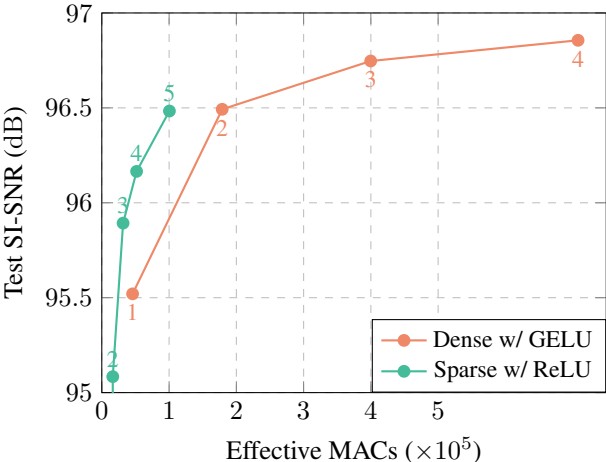

*Figure 9.* Pareto fronts for S5 network test accuracy as a function of effective compute on SpeechCommands V2-35 keyword spotting task. S5 networks with weight and activation sparsity (green) exhibit a domain of Pareto optimality versus dense S5 networks (orange). Number annotations enumerate increasing S5 dimensionality configurations. Further scaling of the sparse architectures would be required to compare with the dense models at higher accuracy.

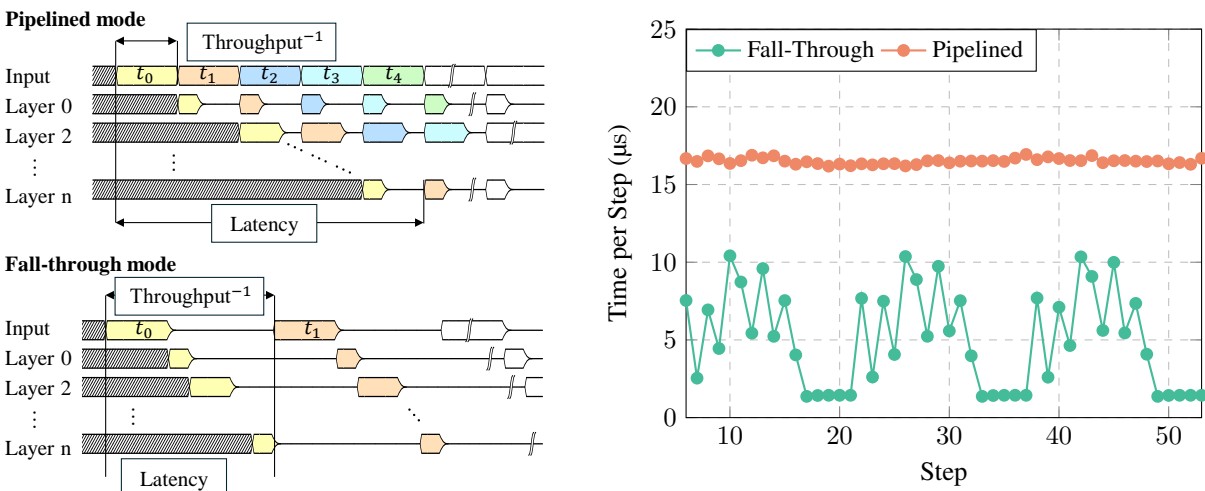

*Figure 10.* (a) Loihi 2 offers two processing modes that optimize either throughput or latency. In the *pipelined mode*, a new data point is inserted in each time step, to use all processing cores and maximize the throughput–at the expense of latency because equal time bins $t_0 = t_1 = \dots$ are enforced. In the *fall-through mode*, a new data points is only provided once the last data point has been fully processed with minimum latency. Only a single neuronal layer is active at any step as data travels through the network. The time per step is thus minimized as traffic is reduced and potentially more complex neuronal layers are not updated. (b) Comparison of execution mode and time per step.

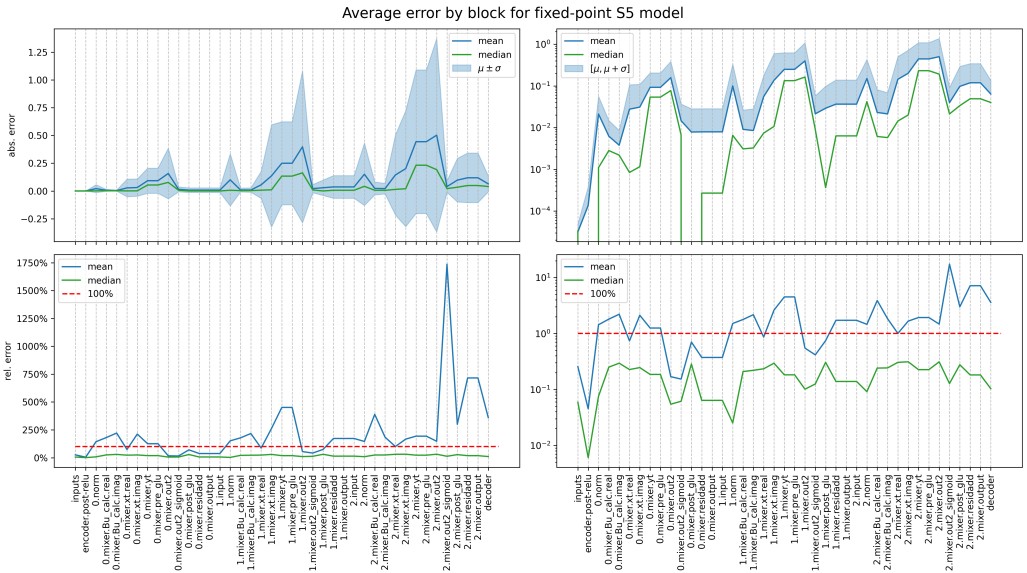

*Figure 11.* Layer-wise analysis of mismatch between the fixed-point model in JAX against the base model using floating-point weights and activations. The **left** and **right** side show the same data with a linear y-axis and log y-axis, respectively. The **top** panels show the mean absolute error $N^{-1} \sum_i^N |x_i - x'_i|$ for all components of the model while the **bottom** panels show the mean relative error $N^{-1} \sum_{\{i \mid i \in \{0,\dots,N\} \wedge x_i \neq 0\}}^N |x_i - x'_i|/|x_i|$. For further explanation, see text.

