# OpenReview forum: "Accelerating Linear Recurrent Neural Networks for the Edge with Unstructured Sparsity"
_ICML.cc/2025/Conference — ICML 2025 poster_

### Official Review · Reviewer_EUuh · 2025-03-08

**Overall Recommendation:** 3

**Summary:**

This paper explores compression techniques for linear RNNs, including unstructured sparsity and fixed-point quantization, and evaluates their acceleration on neuromorphic hardware. The study investigates the trade-offs between latency, energy, and accuracy compared to dense RNNs by introducing these compression techniques. The results demonstrate that highly sparse linear RNNs achieve improved efficiency-performance trade-offs, with 2× less compute and 36% less memory at iso-accuracy. Additionally, quantizing the sparse models and deploying them on the Intel Loihi 2 neuromorphic chip yields significant speed and energy improvements over edge GPUs.

## Update after rebuttal

(Was previously in official comment not visible by the authors)
Thanks for the clarification. It would be beneficial to see the extended results and ablation study in the appendix. For now I will keep the score as 3.

**Claims And Evidence:**

The paper claims that sparse linear RNNs provide better efficiency-performance trade-offs than dense baselines, with notable reductions in computation and memory usage. Furthermore, it states that quantized sparse models on Intel Loihi 2 outperform edge GPUs in terms of speed and energy efficiency. These claims are supported by experimental results, which show improvements in denoising quality while maintaining computational efficiency. The evidence presented in the paper generally aligns with these claims.

**Essential References Not Discussed:**

No.

**Experimental Designs Or Analyses:**

The experimental design is based on the S5 model built with JAX. JAXPruner is used for pruning the network to introduce unstructured sparsity, while the AQT library is used for quantization-aware training. The final network is deployed and evaluated on the Intel Loihi 2 neuromorphic chip. The experimental design appears valid, leveraging appropriate tools and methodologies to assess the impact of sparsity and quantization.

**Methods And Evaluation Criteria:**

The proposed method is evaluated using the Intel Neuromorphic Deep Noise Suppression Challenge, which focuses on human speech denoising. The dataset is derived from the Microsoft DNS Challenge, containing clean human speech and noise source samples. The denoising quality is assessed using the scale-invariant signal-to-noise ratio (SI-SNR), which is an appropriate metric for this application. Given the embedded system context, the chosen evaluation settings are relevant and reasonable.

**Other Comments Or Suggestions:**

* It would be beneficial to evaluate how different linear RNN variants respond to sparsity and quantization to generalize the findings further.
* Additional analysis of various sparse ratios and quantization formats could provide deeper insights into trade-offs and optimal configurations.

**Other Strengths And Weaknesses:**

Strengths:
* The paper successfully demonstrates an improved Pareto optimal front, achieving better denoising quality for the same multiply-accumulate operations (MAC) or reducing MAC while maintaining denoising quality.
* The results show competitive performance compared to an edge GPU (Jetson Orin Nano) in terms of latency and energy efficiency.
Weaknesses:
* The study focuses primarily on a single design point: the S5 model on the Intel Loihi 2 chip. It does not explore how different variants of linear RNNs are affected by sparsity and quantization.
* The paper examines model dimension versus denoising quality but does not sufficiently discuss other critical design parameters such as sparsity ratio and quantization format (beyond W8A16).

**Questions For Authors:**

* Have you considered evaluating the impact of sparsity on other linear RNN architectures beyond the S5 model?
* Can you provide insights into how different quantization formats (other than W8A16) might affect the performance and efficiency of the model?

**Relation To Broader Scientific Literature:**

The paper provides a sufficient literature review covering key topics such as linear RNNs, sparsity, model compression, and neuromorphic computing. The citations are relevant and provide necessary context for understanding the contributions of the work.

**Theoretical Claims:**

The paper does not make any theoretical claims.

---

> ### Author Rebuttal · Authors · 2025-04-01
>
> We gratefully thank the reviewer for their effort and feedback. We would like to address their questions, and we’d appreciate any raise in score if our arguments convince them. We will be happy to hear and address any further feedback.
>
> ### Generalization to different architectures: preliminary results on HGRN
> To expand our results to different linear RNN architectures beyond S5, we can present some preliminary results on training a 370M parameter language model based on the HGRN architecture [1]. Using the same iterative magnitude pruning setup as described in our paper, we trained the HGRN model on the FineWeb dataset for ~5B tokens with different target sparsity ratios from 0% to 90%. The training loss for the 90% sparse model is ~9% higher than for the dense model, see the Table below. Additional details are available in the Figure available at https://figshare.com/s/a8cd30c07af56f4ddbbf. These findings confirm a similar trend as what we reported in our paper on real-time sequence modeling tasks. More work is underway to scale up the language modeling experiments, and verify their validity on real-world datasets and benchmarks.
>
> |                        | Dense | 50%   | 80%   | 90%   |
> |------------------------|-------|-------|-------|-------|
> | Training loss          | 2.68  | 2.72  | 2.82  | 2.90  |
> | Relative loss change   | 0%    | +1.5% | +5.3% | +8.3% |
> | MACs (10^6)            | 341   | 187   | 95    | 64    |
> | Relative MAC change    | 0%    | -45%  | -72%  | -81%  |
>
> ### Ablation study on quantization schemes and sparsity levels
> We agree with the reviewer that exploring the large design space of quantization schemes and sparsity distributions would provide additional insights in this domain. On quantization, previous work on S5 [2] showed that, compared to W8A8, lower precision significantly impacted accuracy. Since Loihi’s message format doesn’t provide any performance advantage for 8-bit versus 16-bit activations, we selected W8A16 which preserves higher accuracy. Regarding sparsity, we selected a 90% target after some initial experimentation, as it resulted in a good trade-off between impact on task accuracy and MACs reduction. We could add an ablation study on sparsity levels to Appendix A, if the Reviewer finds it helpful for readers. We would also acknowledge these directions for future work by updating the last sentences of Section 4 with the following:
>
> *Finally, further exploration of quantization schemes and sparsification techniques could offer deeper insights into optimal model design for different hardware platforms. In particular, leveraging advanced data types for quantization (e.g., FP8) and adopting a more fine-grained selection of sparsity levels, potentially guided by iterative hardware profiling, are promising directions for future research.*
>
> #### References
> 1. Zhen Qin, Songlin Yang, Yiran Zhong, “Hierarchically Gated Recurrent Neural Network for Sequence Modeling”, NeurIPS 2023 Spotlight.
> 2. Abreu, Steven, et al. "Q-S5: Towards quantized state space models." arXiv preprint arXiv:2406.09477 (2024).

---

### Official Review · Reviewer_btkY · 2025-03-13

**Overall Recommendation:** 3

**Summary:**

This paper explores the efficiency-performance trade-offs of sparse linear RNNs through a scaling study. The models achieve SOTA results in real-time audio denoising. By quantizing and deploying them on the Intel Loihi 2 neuromorphic chip, the work significantly reduces latency and energy consumption compared to a dense model on an edge GPU.

**Claims And Evidence:**

Yes

**Essential References Not Discussed:**

No missing essential references were identified.

**Experimental Designs Or Analyses:**

The presentation of Figure 6 lacks clarity. The experimental conditions for Base and QAT are not very clear. Please provide a detailed explanation of what Base represents in each experiment.

**Methods And Evaluation Criteria:**

Yes

**Other Comments Or Suggestions:**

Is FPX in Figure 6 actually intended to be FXP, meaning fixed-point?

**Other Strengths And Weaknesses:**

I think the paper is reasonably clear and well-supported by sufficient evidence. However, its organizational structure could be improved. For example, the section on Compressing Linear RNNs includes a significant amount of related work and background information, which could be streamlined without disrupting the logical flow. Additionally, the discussion of contributions is relatively weak and could be expanded. Furthermore, the section on hardware deployment could provide more details to enhance clarity.

**Questions For Authors:**

Could you provide additional related works on deploying linear RNNs to neuromorphic devices, along with performance comparisons?

**Relation To Broader Scientific Literature:**

This work emphasizes the suitability of linear RNNs for real-time, long-range sequence modeling on edge devices, while noting that their compression and acceleration remain underexplored. However, compression and acceleration of linear RNNs have not been thoroughly explored. Additionally, the paper notes that linear RNNs are a promising match for neuromorphic processors. The study explores sparsification and quantization of linear RNNs and deploys the compressed models on the Intel Loihi 2 neuromorphic chip for real-time processing

**Theoretical Claims:**

Yes, they are correct.

---

> ### Author Rebuttal · Authors · 2025-04-01
>
> We gratefully thank the reviewer for their effort and feedback. We would like to address their questions, and we’d appreciate any raise in score if our arguments convince them. We will be happy to hear and address any further feedback.
>
> ### Clarification on Figure 6
> We regret the suboptimal presentation of Figure 6 and appreciate the reviewer’s nudge to clarify. We confirm that FPX is a typo and should read FXP, for fixed-point. We will replace the caption with the following in the camera-ready version:
>
> *Base: trained a 32-bit floating-point model and applied post-training quantization without additional quantization-aware fine-tuning. QAT: trained the model with quantization-aware training using W8A16 quantization for the forward pass, as straight-through estimators for the backward pass. The results show that the Base model without QAT performs slightly better in FP32 than the QAT model, but significantly worse in static quantization and fixed-point precision. The model shown here is the sparse-6 variant, see Figure 4.*
>
> ### Previous work on deployment of linear RNNs to neuromorphic devices
> Since the now broad interest in linear RNNs is relatively recent, there is limited previous work on deployment of this architecture to neuromorphic devices, and, to the best of our knowledge, no previous work on hardware-aware model compression in this context. Previous work on Loihi 2 demonstrated the implementation of S4D [1], a state space model based on the Legendre Delay Network [2], and a MatMul-free LLM [3].
> While S4D could potentially target the same application domains as S5, Ref. [1] only reports benchmarks on simple synthetic tasks (sMNIST and sCIFAR) and without model compression. For this reason, a direct performance comparison is not possible. Ref. [2] implemented a state space model with a spiking neural network on a neuromorphic processor – also without model compression techniques. The MatMul-free LLM in Ref. [3] uses ternary weight matrices for model compression but doesn’t leverage sparsity advantages from the neuromorphic chip on which it is deployed, thereby making the comparison to our present work difficult.
> We are not aware of any further deployment of linear RNNs on neuromorphic hardware.
>
> #### References
> 1. Meyer, Svea Marie, et al. "A Diagonal Structured State Space Model on Loihi 2 for Efficient Streaming Sequence Processing." arXiv preprint arXiv:2409.15022 (2024).
> 2. Gaurav, Ramashish, Terrence C. Stewart, and Yang Yi. "Legendre-SNN on Loihi-2: Evaluation and Insights." NeurIPS 2024 Workshop Machine Learning with new Compute Paradigms.
> 3. Abreu, Steven, et al. "Neuromorphic Principles for Efficient Large Language Models on Intel Loihi 2." arXiv preprint arXiv:2503.18002 (2025).

---

> > ### Comment · Reviewer_btkY · 2025-04-04
> >
> > Thank the authors for the response. I would like to keep my score.

---

### Official Review · Reviewer_jRFp · 2025-03-14

**Overall Recommendation:** 3

**Summary:**

This paper explores unstructured sparsity in linear recurrent neural networks (RNNs) to improve efficiency in edge AI applications, particularly when deployed on neuromorphic hardware (Intel Loihi 2). The authors examine various model compression techniques and conduct a scaling study to determine the Pareto front of performance vs. efficiency. The study highlights unstructured sparsity as a viable strategy for real-time processing in low-power edge devices.

**Claims And Evidence:**

The claim of the findings in this work can generalize to edge AI beyond neuromorphic hardware is questionable. Only one neuromorphic chip (Loihi 2) and one task (audio denoising) was tested.

**Essential References Not Discussed:**

I am not an expert in either linear recurrent neural networks or neuromorphic computing, so I cannot assess whether any relevant prior findings are missing from the paper.

**Experimental Designs Or Analyses:**

See "Methods And Evaluation Criteria".

**Methods And Evaluation Criteria:**

It makes sense that the performance is measured using Scale-Invariant Signal-to-Noise Ratio (SI-SNR), compute (MACs), memory usage, latency, and energy consumption. Results are compared with dense models running FP32 on a Jetson Orin Nano. However, no evaluation on other neuromorphic processors (e.g., SpiNNaker 2, IBM NorthPole). Only one downstream task is evaluated.

**Other Comments Or Suggestions:**

As a conclusion, I suggest the authors to make a further step in designing original methodology of model compression techniques for neuromorphic hardware in edge. The generalizability of the current results require more comprehensive experiments. The authors should also reconsider if they want to position this work specifically for "neuromorphic hardware" or even more general edge computating, which should be reflected correctly in the paper title.

**Other Strengths And Weaknesses:**

Strengths:
1. The investigation of deep learning model compression in neuromorphic computing has substantial real-world impact.
2. The on-device experiment results are very beneficial for the community regarding edge AI deployment.

Weaknesses:
1. This work is more like an experimental report rather than an academic paper with solid intellectual contribution. The model and the compression methods are all off-the-shelf.
2. The profiling of model compression on neuromorphic hardware is only conducted on a single device model (Intel Loihi 2), which weakly support the conclusion from the experimental results.
3. Only a single task (audio denoising) is evaluated. A broader range of tasks need to be assessed to sufficiently support the conclusion for the general edge AI applications.

**Questions For Authors:**

1. Why was only Intel Loihi 2 used? Would the results generalize to other neuromorphic chips like SpiNNaker 2 or IBM NorthPole?
2. Would the observed efficiency trends hold for other real-time tasks (e.g., NLP, speech recognition, or perception)?
3. How does batch size impact efficiency? Could Loihi 2 be optimized for batch processing?

**Relation To Broader Scientific Literature:**

The paper builds on prior work in structured state-space models (S5), model compression, and neuromorphic hardware. The experimental results in this work can be good references for the deployment of linear RNN models on edge devices.

**Theoretical Claims:**

No theoretical claims are described in the paper.

---

> ### Author Rebuttal · Authors · 2025-04-01
>
> We gratefully thank the reviewer for their effort and feedback. We would like to address their questions, and we’d appreciate any raise in score if our arguments convince them. We will be happy to hear and address any further feedback.
>
> ### Generalization to different accelerators
> Please refer to our response to Reviewer VTiJ.
>
> ### Generalization to different tasks
> Please refer to our response to Reviewer VTiJ.
>
> ### Impact of batch processing
> We appreciate the reviewer’s question regarding the impact of batching on our benchmarking results. We address it in the following paragraph, which we plan to include in Section 3.3 of the camera-ready version.
>
> *Although edge applications are typically thought of as single-batch applications, some workloads at the edge require small-batch inference, e.g., de-noising audio streams from multiple on-device microphones. For this reason, it is important to investigate how batch processing affects latency and energy efficiency for the two hardware architectures. While Intel Loihi 2 doesn’t natively support batching in the sense of processing multiple independent samples though the same model instantiation, the parallel inference of independent sequences can be obtained by replicating the model on the chip as many times as required by batch size. We extended the results in Table 1 to compare the effect of this implementation of batching to the usual batch processing of the Jetson Orin GPU. Figure X (available at https://figshare.com/s/4d6035c9a2c4739d7201) shows total latency and energy per sample across batch sizes, from 1 to 32. The results demonstrate that our approach on Loihi 2 maintains its large latency advantage while the energy efficiency gain is only maintained for the small-batch regime (below 8 samples), which is typical of edge inference applications. As expected, batch processing improves energy efficiency for the GPU, since the cost of data movement associated with loading the model is offset by the parallelized evaluation of multiple samples.*
>
> ### Novelty and originality of the work
> We thank the reviewer for pointing out that we missed to express the novelty and originality of our work. We also appreciate their statement that our work “can be a good reference for the deployment of linear RNN models on edge devices”. In addition, we would argue that our paper has two further main original/novel contributions.
>
> First, this paper spearheads the idea that neuromorphic processors are an ideal platform for the emerging class of linear RNNs. This is particularly due to the tight integration of massively parallel compute and memory in neuromorphic hardware, which can efficiently update stateful recurrent neurons. Neuromorphic processors are typically designed for low-latency processing of sequentially incoming sensory signals, and can thus particularly benefit from the advantageous scaling trends of linear RNNs with long sequences. Our present paper is the first peer-reviewed publication that explores the combination of model compression techniques required to exploit the synergy between neuromorphic processors and linear RNNs.
>
> Second, while we agree that we applied off-the-shelf compression techniques, we would argue that our paper combines them into a novel training recipe necessary to leverage the specific features typical for neuromorphic computing for an optimized deployment of linear RNNs. We would like to kindly point out that, according to the 2025 ICML reviewer guidelines, “Originality need not mean wholly novel methods. It may mean a novel combination of existing methods to solve the task at hand, a novel dataset, or a new way of framing tasks or evaluating performance so as to match the needs of the user.” Our new training and deployment recipe has proven to bring tangible performance and energy advantages on real hardware for a real-world task that requires low-latency, low-power execution, audio denoising. We will release this new training pipeline upon acceptance so that the community can extend it or apply it to additional applications.
>
> We will update the introduction in the camera-ready version accordingly.

---

> > ### Comment · Reviewer_jRFp · 2025-04-01
> >
> > Thanks the authors for the response. My previous conerns are mostly addressed. I would like to raise my rating to a 3.

---

### Official Review · Reviewer_VTiJ · 2025-03-17

**Overall Recommendation:** 4

**Summary:**

The paper presents a method to accelerate the computations of linear Recurrent Neural Networks (RNNs) using unstructured sparsity for edge computing applications. This work is motivated by a case study showing that highly sparse linear RNNs achieve superior efficiency-performance trade-offs compared to dense baselines. The paper particularly highlights the deployment of sparse linear RNNs on the Intel Loihi 2 neuromorphic processor, where quantized models demonstrate significant reductions in latency (~42×) and energy consumption (~149×) compared to edge GPUs.

**Claims And Evidence:**

The paper supports its claims about the efficiency and performance benefits of unstructured sparsity in linear RNNs by reporting its implementation results from the Intel Loihi 2 chip. The results demonstrate significant reductions in latency (42×) and energy consumption (149×) compared to an edge GPU, supported by benchmarks on real-world tasks like audio denoising.

**Essential References Not Discussed:**

N/A

**Experimental Designs Or Analyses:**

Yes, the experimental design appears OK specially in evaluating sparsity effects, quantization, and hardware acceleration on Loihi 2 vs. Jetson Orin Nano. The Pareto front analysis shows interesting trade-offs between accuracy, compute efficiency, and memory.

**Methods And Evaluation Criteria:**

Yes, the proposed methods and evaluation criteria align well with the problem of accelerating linear RNNs for edge applications. The Intel N-DNS Challenge benchmark is a good test case for audio denoising, and the comparison between sparse vs. dense models on Loihi 2 vs. Jetson Orin Nano provides is interesting.

**Other Comments Or Suggestions:**

N/A

**Other Strengths And Weaknesses:**

Strengths:

-- Low-latency, memory-efficient implementation of AI models is highly important for real-time streaming applications, which motivates the research conducted in this paper.

-- The authors empirically validate their claims by studying Pareto efficiency trade-offs across different compute budgets.

-- The integration of neuromorphic processors for accelerating sparse models is very interesting. It provides practical results which supports their simulation results.


Weaknesses:

-- While Loihi 2 is well-suited for sparse and event-driven computations, it remains unclear how these optimizations would translate to conventional customized hardware. How does it compare to customized accelerators for sparse or quantized models?

-- The focus of this work is on audio denoising. How does the proposed method could generalize to other domains (e.g., NLP and vision applications).

**Questions For Authors:**

See my points listed as weaknesses.

**Relation To Broader Scientific Literature:**

The paper builds on prior works in sparse models and neuromorphic computing. The main broader impact of this work is its validation on neuromorphic hardware for streaming tasks, pushing the field toward practical edge AI deployment.

**Theoretical Claims:**

There is not much theory in this work.

---

> ### Author Rebuttal · Authors · 2025-04-01
>
> We gratefully thank the reviewer for their effort and feedback, and we would like to address their questions. We will be happy to hear and address any further feedback.
>
> ### Generalization to different accelerators
> Reviewers VTiJ and jRFp raised the need for a discussion about the generalizability of our methodology and benchmarking to other hardware platforms (e.g., SpiNNaker 2 and IBM NorthPole). While we adopted a hardware-aware approach focused on Loihi 2, we believe that platforms with similar feature sets can benefit from it. However, since most neuromorphic processors are currently research prototypes, they don’t provide public access (like IBM NorthPole) and/or lack a high-level programming framework that would allow the transfer of models from one platform to another. Following the approach proposed in NeuroBench [1], we will publicly release our training pipeline and checkpoints to enable the implementation on other neuromorphic processors by the relevant experts.
>
> We further address the generalizability discussion in the following paragraph, which we plan to add to Section 4 in the camera-ready version.
>
> *While the proposed hardware-aware methodology is tailored to leverage the Loihi-specific feature set, we believe that platforms with similar characteristics could potentially benefit from the results we presented. Neuromorphic processors such as SpiNNaker 2 [2] and IBM NorthPole [3], which presents similar architecture patterns to Loihi 2, would greatly benefit from our methodology on activation and weight sparsity. In addition, other platforms with tight compute-memory integration, such as Cerebras Wafer-Scale Engine (WSE-3) [4], provide support for unstructured sparsity, even if targeting datacenter-scale applications.*
>
> ### Generalization to different tasks
> Reviewers VTiJ and jRFp noted that extending our benchmarking to other datasets would strengthen our claims and the relevance of our work for the broader edge inference community. Starting from the S5 baseline [5] and following our methodology, we are running experiments on the keyword spotting SpeechCommands V2-35 (SC35) dataset [6], which provides a common real-time use case for the edge. The preliminary results, available at https://figshare.com/s/3d0ba8e3f515535871ba, compare the training curves for a narrow dense and wider sparse model (90% sparse weights and ReLU activations) with similar inference MACs. The plot shows that the sparse model (still under training) is on track to match or exceed the accuracy of the dense counterpart. Similar experiments are underway across inference compute budgets, and we plan to add such results in the same form as Figure 4 to the camera-ready version. Since our methodology seems to generalize without modification to SC35, we expect it to also generalize to other sequence modeling applications commonly solved with linear RNNs, such as bio-marker signal monitoring [7], time series forecasting [8], or action recognition [9].
>
> #### References
> 1. Yik, Jason, et al. "The neurobench framework for benchmarking neuromorphic computing algorithms and systems." Nature Communications 16.1 (2025): 1545.
> 2. Mayr, Christian, Sebastian Hoeppner, and Steve Furber. "SpiNNaker 2: A 10 million core processor system for brain simulation and machine learning-keynote presentation." Communicating Process Architectures 2017 & 2018. IOS Press, 2019. 277-280.
> 3. Modha, Dharmendra S., et al. "Neural inference at the frontier of energy, space, and time." Science 382.6668 (2023): 329-335.
> 4. Lie, Sean. "Cerebras architecture deep dive: First look inside the hardware/software co-design for deep learning." IEEE Micro 43.3 (2023): 18-30.
> 5. Smith, Jimmy TH, Andrew Warrington, and Scott W. Linderman. "Simplified state space layers for sequence modeling." arXiv preprint arXiv:2208.04933 (2022).
> 6. Warden, Pete. "Speech commands: A dataset for limited-vocabulary speech recognition." arXiv preprint arXiv:1804.03209 (2018).
> 7. Pimentel, Marco AF, et al. "Toward a robust estimation of respiratory rate from pulse oximeters." IEEE Transactions on Biomedical Engineering 64.8 (2016): 1914-1923.
> 8. Schirmer, Mona, et al. "Modeling irregular time series with continuous recurrent units." International conference on machine learning. PMLR, 2022.
> 9. Kuehne, Hildegard, et al. "HMDB: a large video database for human motion recognition." 2011 International conference on computer vision. IEEE, 2011.

---

### Decision · Program_Chairs · 2025-05-01

**Decision:**

Accept (poster)

**Comment:**

This paper studies the Pareto frontier between performance and efficiency for unstructured sparsity in recurrent neural networks at the edge. Using the Intel Loihi 2 neuromorphic chip and quantization of the studied models, this translates even to an 149x lower energy consumption than dense neural networks. All reviewers found the work insightful, with possible real-world impact. After the rebuttal period, they unanimously lean towards the paper acceptance. Consequently, the AC suggests acceptance and recommends to the authors to include all rebuttal discussions and results in the final version of the paper.